# Advances in PCL, PLA, and PLGA-Based Technologies for Anticancer Drug Delivery

**DOI:** 10.3390/pharmaceutics17101354

**Published:** 2025-10-20

**Authors:** Yeongbeom Kim, Jaewoo Kwak, Minyeong Lim, Su Yeon Lim, Sehyun Chae, Suk-Jin Ha, Young-Wook Won, Hyun-Ouk Kim, Kwang Suk Lim

**Affiliations:** 1Department of Smart Health Science and Technology, Kangwon National University, Chuncheon 24341, Republic of Korea; kyb8234@kangwon.ac.kr (Y.K.); wodn332@kangwon.ac.kr (J.K.); alsdudahs@kangwon.ac.kr (M.L.); suyeonlim9846@kangwon.ac.kr (S.Y.L.); shchae@kangwon.ac.kr (S.C.); sjha@kangwon.ac.kr (S.-J.H.); 2Department of Biotechnology and Bioengineering, College of ACE, Kangwon National University, Chuncheon 24341, Republic of Korea; 3Institute of Fermentation of Brewing, Kangwon National University, Chuncheon 24341, Republic of Korea; 4Department of Biomedical Engineering, College of Engineering, University of North Texas, Denton, TX 76203-5017, USA

**Keywords:** polycaprolactone (PCL), polylactic acid (PLA), poly(lactic-*co*-glycolic acid) (PLGA), biodegradable polymers, anticancer drug delivery

## Abstract

Biodegradable polymers such as Polycaprolactone (PCL), Polylactic acid (PLA), and Poly(lactic-*co*-glycolic acid) (PLGA) are attracting attention as key platforms for anticancer drug delivery systems due to their excellent biocompatibility and controllable degradation rates. These polymers can overcome limitations of existing chemotherapeutics, such as low bioavailability, systemic toxicity, and nonspecific cell damage, and contribute to the development of precision medicine approaches and long-acting therapeutics. This paper discusses the chemical and physicochemical properties of these three polymers, their synthetic strategies, and the controlled drug release technology through surface functionalization and stimuli-responsive design. Furthermore, we highlight their potential for use in various formulations, including micelles, nanoparticles, hydrogels, and microspheres, enabling enhanced drug solubility, sustained release, and tumor targeting. Preclinical and clinical applications demonstrate that these polymer-based DDSs represent a promising approach for implementing next-generation precision anticancer treatment strategies, with further potential for clinical translation and widespread adoption.

## 1. Introduction

Cancer remains a serious global health problem, and existing chemotherapy regimens have clear limitations due to nonspecific drug distribution and systemic toxicity. Furthermore, many anticancer drugs have low water solubility and a narrow therapeutic index, resulting in limited efficacy and serious side effects. Therefore, drug delivery systems (DDSs) have emerged as a key strategy for improving drug solubility and stability, precisely controlling the rate and location of drug release, and selectively delivering drugs to target tissues [1]. Polymer-based DDSs have been developed in various forms, including nanoparticles, micelles, hydrogels, microspheres, and drug-polymer conjugates, and can maintain drug concentrations in the body for extended periods, minimizing damage to normal tissues, and maximizing tumor accumulation [2]. In particular, polymer nanoparticles can be surface-modified to enhance active targeting and in vivo stability, and their size and charge characteristics can be tailored to exploit the vascular permeability and retention (EPR) effect of tumor tissue. These properties lead to precision medicine approaches that increase treatment efficacy and reduce side effects compared to conventional chemotherapy [3].

Among these, biodegradable polymers ensure long-term safety because they are broken down and excreted into harmless metabolites in the body. Furthermore, they do not require surgical removal, making them ideal materials for implementing DDS [1]. Representative synthetic aliphatic polyesters include polycaprolactone (PCL), polylactic acid (PLA), and poly(lactic-*co*-glycolic acid) (PLGA). All of these polymers exhibit excellent biocompatibility and differ in physical and chemical properties and degradation rates, allowing them to be tailored to various therapeutic strategies [4]. For example, PCL is suitable for extended-release formulations lasting several months or longer due to its high hydrophobicity and slow degradation rate [5]. PLA offers a controlled release period ranging from several days to several months, depending on its molecular weight and crystallinity, and has been studied in various parenteral formulations [6]. On the other hand, PLGA, with its lactic acid:glycolic acid (LA:GA) ratio, allows for precise control of degradation rate and hydrophilicity, making it one of the most widely utilized polymers for finely designing drug release patterns [7].

These polymer-based DDSs play a crucial role in overcoming fundamental limitations of existing anticancer chemotherapy. Encapsulating anticancer drugs in polymer matrices extends their in vivo stability and half-life, enabling selective and sustained release at the tumor site [8]. This allows for maintaining intratumoral drug concentrations while minimizing exposure to normal tissues and alleviating drug resistance [9]. Indeed, PLA/PLGA-based formulations have significantly improved the solubility of poorly soluble drugs such as paclitaxel and doxorubicin, progressing to clinical trials and approval stages. PCL-based formulations are also showing promise in therapeutic strategies requiring ultra-long-acting drug release [10]. Despite this, existing review articles often focus on specific polymers (PCL or PLGA) or specific formulations, or provide an overview of DDSs without sufficiently addressing their connection to anticancer therapy. This review addresses these limitations by examining the three polymers, PCL, PLA, and PLGA, from a comparative and integrated perspective. By comprehensively covering the chemical and physical properties, synthetic strategies, functionalization techniques, drug release mechanisms, formulation types, and preclinical and clinical applications of the three polymers, it aims to provide readers with systematic criteria for selecting a suitable polymer for a specific therapeutic goal [11,12]. Furthermore, by analyzing recent research achievements and FDA approval cases, it differentiates itself from previous reviews by going beyond a simple technological overview to highlight their potential for clinical translation and their significance as precision anticancer treatment platforms [13].

Therefore, by comprehensively comparing these three representative biodegradable polymers, this review aims to provide in-depth insights into the design principles and clinical applicability of polymer-based anticancer DDSs. This aims to provide a basis for researchers and clinicians to select and design optimal polymer carriers and to contribute to the development of precision medicine anticancer treatment strategies in the future.

## 2. Structural and Physicochemical Properties of PCL, PLA, and PLGA

### 2.1. Chemical Composition and Biodegradability

Polycaprolactone (PCL), polylactic acid (PLA), and poly(lactic-*co*-glycolic acid) (PLGA) are all aliphatic polyester polymers containing ester bonds, exhibiting unique biodegradation behaviors depending on their chemical composition and microstructure (Figure 1).

PCL is a semi-crystalline polymer synthesized by ring-opening polymerization of ε-caprolactone monomers. Due to its high crystallinity and hydrophobicity, PCL exhibits a slow rate of pure hydrolysis, and complete biodegradation can take years. Degradation occurs gradually, primarily through the action of microbial enzymes or chemical hydrolysis in the presence of moisture [14] (Table 1).

PLA is composed of lactic acid derivatives, and its crystallinity and physical properties vary significantly depending on the ratio of the D- and L-isomers. Low D-isomer content exhibits high crystallinity, while high D-isomer content results in an amorphous structure. In the presence of moisture, PLA degrades into lactic acid monomers through hydrolysis of ester bonds, which are then completely converted to carbon dioxide and water through the body’s metabolic pathways [15] (Table 1).

PLGA is a copolymer of PLA and PGA, and its degradation rate and physical stability are controlled by the molar ratio of lactic acid (LA) to glycolic acid (GA). Typically, a 50:50 LA:GA ratio results in the fastest degradation, and both the generated lactic acid and glycolic acid are ultimately excreted harmlessly through the body’s metabolic pathways [16] (Table 1).

### 2.2. Thermal and Mechanical Properties

The thermal and mechanical properties of these polymers directly influence not only processability but also drug release rates and stability.

PCL has a relatively low melting point (approximately 58–61 °C) and a very low glass transition temperature (T_g ≈ −60 °C). The degree of crystallinity varies within a range of approximately 20–33% depending on processing conditions. Higher crystallinity increases the heat deflection temperature and elastic modulus, leading to increased strength. However, excessive injection flow rates can reduce crystallinity and thus thermal resistance [17].

PLA has a relatively high T_g of approximately 60 °C and melting point of 150–160 °C, and exhibits robust properties with a tensile strength of 50–70 MPa and an elastic modulus of 3–4 GPa. However, prolonged exposure to high temperatures and shear during melt processing leads to chain scission, which reduces molecular weight and consequently degrades thermal stability and mechanical properties. Insufficient crystallinity can lead to reduced strength and stiffness, leading to material brittleness [18].

The thermal properties of PLGA are controlled by the LA/GA ratio and molecular weight, with a T_g typically ranging from 40 to 60 °C. Higher lactide content improves T_g and structural stability, but higher Glycolide content increases hydrophilicity, leading to accelerated degradation and loss of mechanical integrity. Furthermore, processing processes involving heat and shear, such as hot melt extrusion or 3D printing, can result in a decrease in molecular weight and deterioration of physical properties [19].

### 2.3. Degradation Mechanisms and Their Impact on Drug Release Kinetics

Chemical composition and thermomechanical properties are closely linked to degradation behavior in the biological environment, which directly influences the rate and pattern of drug release. All three polymers degrade through hydrolysis of ester bonds, but exhibit different rates and patterns of degradation due to differences in crystallinity, hydrophobicity, and copolymerization ratio.

PCL exhibits a very slow degradation rate due to its high crystallinity and hydrophobicity, which limits water penetration. Its long-term structural stability suppresses initial burst release and enables consistent sustained release [20] (Figure 2).

PLA primarily degrades via non-enzymatic hydrolysis, and its degradation rate varies depending on crystallinity, molecular weight, and processing history. Highly crystalline and high molecular weight PLA can persist in the body for months or longer, whereas amorphous PLA can completely degrade within weeks. The release period can be controlled from days to months through composition and processing conditions [21] (Figure 2). PLGA’s hydrophilicity and degradation rate are precisely controlled by varying the LA:GA ratio. Higher glycolide content leads to faster degradation, while higher lactide content favors sustained release. Drug hydrophilicity and hydrophobicity also significantly influence the release pattern. Hydrophilic drugs induce rapid dissolution and accelerated polymer degradation, whereas hydrophobic drugs retain the matrix for extended periods and exhibit gradual release [12] (Figure 2).

Thus, the degradation characteristics of PCL, PLA, and PLGA serve as key variables in the design and performance optimization of drug delivery systems. Process strategies, including formulation design, surface modification, and functional group introduction, must be implemented in tandem to ensure optimal therapeutic duration and release profiles.

A summary of the key findings described in Section 2.1, Section 2.2 and Section 2.3 is presented in Table 1.

**Table 1 pharmaceutics-17-01354-t001:** Comparison of PCL, PLA, and PLGA Properties.

Polymer	PCL	PLA	PLGA
Chemical Composition	Semi-crystalline aliphatic polyester; synthesized by ROP of ε-caprolactone.	Aliphatic polyester; derived from L-lactide and D-lactide (chiral isomers).	Copolymer of lactic acid (LA) and glycolic acid (GA); tunable LA:GA ratio.
Crystallinity	20–33% (high crystallinity).	Varies by D/L isomer ratio; low D = crystalline, high D = amorphous.	Amorphous to semi-crystalline depending on LA:GA ratio.
Melting Point	58–61 °C	150–160 °C	Not well-defined; varies by LA:GA ratio, typically amorphous (no sharp Tm).
Glass Transition	≈−60 °C	≈60 °C	40–60 °C (higher LA → higher Tg).
Mechanical Properties	Flexible; low tensile strength; strength increases with crystallinity.	Tensile strength: 50–70 MPa; Elastic modulus: 3–4 GPa; brittle if low crystallinity.	Properties depend on LA:GA; higher LA = more rigid, higher GA = more hydrophilic and weaker.
Biodegradation Behavior	Very slow hydrolytic degradation; complete biodegradation may take years.	Hydrolytic degradation into lactic acid → metabolized to CO_2_ + H_2_O.	Hydrolytic degradation; fastest at 50:50 LA:GA ratio; metabolites excreted safely.
Degradation Rate	Slowest (months–years).	Intermediate (weeks–months depending on crystallinity/MW).	Tunable (days–months depending on LA:GA ratio, 50:50 degrades fastest).
Drug Release Characteristics	Minimal burst release; stable zero-order release; good for long-term implants.	Three-stage release profile; tunable from days to months.	Highly tunable; hydrophilic drugs accelerate degradation, hydrophobic drugs prolong release.

## 3. Synthesis and Functionalization of PCL, PLA, and PLGA for Drug Delivery

### 3.1. Ring-Opening Polymerization and Copolymerization Strategies

Ring-opening polymerization (ROP) is widely used to synthesize high-molecular-weight biodegradable polymers [22]. ROP is a polymerization method that induces chain growth through ring-opening of cyclic monomers, and has various reaction pathways, including radical, anionic, cationic, enzyme-mediated, and coordination-based [23]. In particular, PLA is synthesized through ring-opening polymerization of lactide, typically using a tin oxide (Sn(Oct)_2_)-based catalyst [24]. The polymerization yield and final polymer properties are determined by the monomer-to-catalyst ratio, reaction temperature and time, and catalyst type. Side reactions, such as backbiting and transesterification, can lead to non-uniform molecular weight and molecular weight distribution [25,26]. To minimize these side reactions and achieve controlled polymerization, the use of high-purity monomers, removal of moisture from the reaction system (e.g., through reduced-pressure distillation), and the appropriate combination of initiator and catalyst are essential [27]. In particular, the combination of alcohol initiators and metal catalysts has been evaluated as a method for stably producing polymers with desired molecular weights [28]. Recently, the use of organic catalysts or low-toxicity metal catalysts has been attracting attention due to biocompatibility and residual toxicity, which are advantageous for the design of high-purity, functional polymers for drug delivery [29].

### 3.2. Surface Modifications and Functional Group Incorporation

Despite their biodegradability, polymer nanoparticles based on PLA, PLGA, and PCL have limitations such as rapid clearance from the body, induction of immune responses, and nonspecific distribution [30]. To overcome these limitations, strategies for enhancing biocompatibility and imparting targeting properties through surface modification are actively being studied. For example, PEG coating extends blood circulation time and imparts stealth by inhibiting acid binding, while chitosan- and lectin-based coatings promote mucosal adhesion and receptor-mediated uptake [31,32].

Furthermore, the introduction of functional groups such as -COOH, -NH_2_, and -SH provides covalent binding sites for biomolecules, enabling the stable immobilization of targeting ligands such as antibodies, peptides, and aptamers [33,34]. Various surface functionalization techniques, such as EDC/NHS coupling, dopamine self-polymerization, and tannic acid-based self-assembly, are being utilized for this purpose. The physicochemical properties of nanoparticles, such as surface charge, hydrophobicity/hydrophilicity, and mucosal permeability, directly impact cellular uptake and drug delivery efficiency, making precise surface engineering essential [35]. In particular, PCL’s high hydrophobicity limits the introduction of surface functional groups, necessitating surface activation or hydrophilic film formation [36]. In contrast, PLGA and PLA are readily amenable to the introduction of reactive functional groups, making them advantageous for bioactive molecule immobilization and targeting [34,37].

### 3.3. Smart Polymer Design for Controlled Drug Release

Smart polymers, or stimuli-responsive polymers, are polymers that can change their physical and chemical properties, such as solubility, hydrophilicity, structure, and degradability, in response to external stimuli such as temperature, pH, redox state, and enzyme activity. This allows for precise control of the timing and rate of drug release, drawing attention in drug delivery technology [38,39].

Smart polymers can be manufactured from synthetic polymers such as PEG, PLA, PLGA, and PCL, as well as natural polymers such as chitosan, gelatin, and albumin. They offer advantages such as high stability, ease of mass production, and the potential for diverse biocompatibility modifications [40]. Smart polymers actively control drug release by sensing environmental stimuli. Specifically, in anticancer treatment, they respond to specific conditions (pH, enzymes, redox, etc.) of the tumor microenvironment, selectively releasing drugs, thereby enhancing therapeutic efficacy and reducing systemic toxicity [38]. Furthermore, not only single-stimulus but also dual- and multi-stimulus-responsive systems can operate precisely even in complex physiological environments, making them promising for the treatment of refractory tumors [41]. Combining these stimuli-responsive nanoparticles with immunotherapeutic agents can contribute to the induction of long-term immune memory and the prevention of tumor recurrence, while simultaneously improving the nonspecific distribution and rapid drug metabolism of existing treatments [42]. Therefore, introducing smart design into PCL, PLA, and PLGA-based nanoparticles is considered a key strategy to enhance the selectivity, therapeutic durability, and in vivo efficacy of anticancer drug delivery systems [43].

Collectively, precise control of polymer-level polymerization strategies, surface functionalization, and stimuli-responsive design, provides a crucial foundation for maximizing the therapeutic efficacy and biocompatibility of PCL-, PLA-, and PLGA-based drug delivery systems.

## 4. Nano- and Micro-Scale Drug Delivery Systems Based on PCL, PLA, and PLGA

### 4.1. Polymeric Micelles: Self-Assembly and Drug Encapsulation Efficiency

Functionalized polymers such as PCL, PLA, and PLGA, synthesized through polymer synthesis and surface modification, can self-assemble into nanoscale structures, forming unique nanoparticles called micelles [44]. Specifically, the physical interaction between the hydrophobic and hydrophilic moieties of block copolymers spontaneously forms stable micelles in aqueous solutions [45]. These micellar structures enhance drug solubility and prevent drug degradation in the body, significantly enhancing drug encapsulation efficiency [46].

PCL-based micelles offer high drug encapsulation efficiency based on their strong affinity for hydrophobic drugs. PCL-PEG copolymer-based micelles stably encapsulate hydrophobic drugs in their core, while the hydrophilic PEG layer on the outer surface increases the micelle’s in vivo stability and blood circulation time. This allows them to deliver long-term, sustained drug release profiles for various therapeutics, including cancer treatment [47]. PLA-based micelles are widely used, particularly for anticancer drug delivery, due to their rapid and predictable degradation characteristics. Micelles composed of PLA-PEG block copolymers facilitate the introduction of cell-targeting ligands through surface modification, and their combination of high biocompatibility and hydrolytic properties enables efficient and selective intracellular delivery. Furthermore, lactic acid generated from PLA degradation is readily eliminated from the body, minimizing toxicity [48].

PLGA-based micelles offer the advantage of precisely adjusting the monomer composition ratio to achieve a wide range of control over the rate and duration of drug release. Specifically, PLGA-PEG-based micelles exhibit rapid hydrolysis and offer a tunable release profile in vivo, suppressing burst release and maintaining consistent drug concentrations over time. These micelles can be tailored to achieve optimal drug release rates depending on the therapeutic objective and conditions, making them effective in the treatment of various diseases [49]. Thus, micellar structures formed by synthetic and functionalized polymers are establishing themselves as powerful platforms in the field of drug delivery to maximize drug solubility, stability, selective targeting, and in vivo delivery efficiency.

A representative study reported micelles prepared by free radical polymerization of star-shaped PCL/PEG (SSMPEG-PCL) copolymers. This structure maintained high stability based on a low critical micelle concentration (CMC), and exhibited excellent dosorubicin (Dox) anticancer efficacy both in vitro and in vivo, along with pH-dependent drug release [50]. Furthermore, PLA–PEG–Fol micelles modified with folic acid (Fol) increased the solubility and intracellular uptake of curcumin (CUR), thereby enhancing its targeting and cytotoxicity against HepG2 cells [51]. Meanwhile, PLGA–PEG–PLGA triblock copolymer micelles prepared by a solvent-dialysis method effectively improved drug biodistribution by extending drug loading efficiency and blood half-life and reducing nonspecific capture by the reticuloendothelial system (RES) [52] (Table 2).

### 4.2. Nanoparticles and Nanocapsules: Passive vs. Active Targeting Approaches

While polymeric micelles primarily entrap and deliver drugs through the self-assembly of block copolymers, nanoparticle and nanocapsule systems have evolved into platforms that enable more precise control of drug delivery and release through more sophisticated structural design and surface modification. Nanoparticles and nanocapsules typically measure less than a few hundred nanometers and can be used for passive and active targeting strategies, enabling concentrated drug delivery to specific lesion sites [53,54].

PCL-based nanoparticles, with their high hydrophobicity and exceptional structural stability, can effectively utilize the Enhanced Permeability and Retention (EPR) effect, which allows them to passively accumulate in tumor tissues while circulating in the blood for extended periods. Furthermore, surface modification, incorporating specific ligands such as peptides, allows for selective binding to specific receptors on the tumor cell surface, enabling active targeting. This enhances drug therapeutic efficacy and minimizes toxicity in normal tissues [55,56]. PLA-based nanoparticles are biodegradable and biocompatible, and their rapid hydrolysis allows for rapid drug release at the target site. In particular, PLA nanoparticles readily conjugate targeting ligands, such as antibodies or peptides, to their surface, enabling highly efficient selective drug delivery to cancer cells that overexpress specific receptors, such as HER2 or EGFR. This active targeting contributes to enhanced cancer cell specificity and minimizing systemic drug side effects [57,58].

PLGA-based nanocapsules possess a unique structure with separate internal and external structures, allowing for stable encapsulation of liquid drugs or active substances. PLGA nanocapsules can precisely control drug release rates by adjusting their degradation rate, and the introduction of various functional groups on their surface can induce specific interactions with specific cell receptors. This functionalization helps maximize therapeutic efficacy by simultaneously optimizing passive EPR effects and active targeting [59,60]. In conclusion, nanoparticle and nanocapsule systems are attracting attention as drug delivery technologies that can effectively deliver drugs to target sites through more precise and diverse targeting strategies compared to micellar systems, while minimizing side effects in normal tissues.

A representative study reported nanoparticles prepared using ultrasonic emulsification and solvent evaporation using a PCL–PEG–PCL triblock copolymer. This formulation exhibited a particle size of approximately 60 nm and a high encapsulation efficiency of approximately 95%, and exhibited sustained curcumin (CUR) release for up to 96 h, along with an AUC more than four times higher than free CUR and a prolonged blood circulation time [61]. In addition, PEG-PTX/PTX hybrid nanoparticles simultaneously demonstrated improved stability and tumor accumulation rate, potent antitumor efficacy, and low toxicity compared to PEG-PLA/PTX micelles and Taxol® [62]. In addition, PLGA-based paclitaxel (PTX) nanoparticles prepared by a nanoprecipitation method had a spherical particle shape of less than 200 nm and a drug loading efficiency of approximately 90%. They also exhibited a two-stage release pattern combining initial rapid release and sustained release, demonstrating superior cytotoxicity compared to Taxol®, confirming their applicability as an intravenous formulation [63] (Table 3).

### 4.3. Hydrogels and Scaffolds: Applications in Localized Drug Delivery

While the previously discussed nanoparticles and nanocapsules are suitable for selective drug delivery to desired target sites via systemic circulation, hydrogels and scaffolds are larger-scale drug delivery systems that can deliver drugs to specific, stable, and long-term sites, or even serve as scaffolds for tissue regeneration and restoration [64]. Hydrogels are gels formed by three-dimensional cross-linking of hydrophilic polymer networks. Under physiological conditions, they contain a large amount of water and are soft and flexible [65]. Conversely, scaffolds, with their porous structures, provide structural stability that promotes cell penetration and tissue regeneration, while also providing the ability to stably encapsulate and release drugs over a long period of time [66].

PCL-based hydrogels and scaffolds, in particular, offer both prolonged drug release and excellent flexibility, making them widely used in tissue engineering and wound healing [67]. The slow biodegradability of PCL allows for localized, slow drug release, providing long-term, sustained therapeutic effects. Its excellent flexibility facilitates application in biological environments requiring flexibility, such as skin and soft tissues [68].

PLA-based hydrogels and scaffolds are ideal for relatively rapid drug release due to their high mechanical strength and rapid, predictable degradation rates, while simultaneously providing the initial structural support necessary for tissue regeneration and restoration [69]. Specifically, PLA hydrogels degrade along with drug release, supplying biocompatible lactic acid to the localized area, which promotes rapid regeneration of surrounding tissues [70].

PLGA-based scaffolds can precisely control drug release rates by adjusting their composition and porous structure, providing an optimal environment for cell growth and tissue regeneration [71]. Furthermore, PLGA releases acidic byproducts during degradation. Careful design of PLGA scaffolds, which carefully consider drug release and tissue regeneration rates, can maintain an appropriate pH environment in the surrounding tissue and maximize therapeutic efficacy [72]. Thus, hydrogels and scaffolds are larger than nanoscale particle systems and combine structural functions that promote tissue regeneration, making them essential platforms for a variety of medical applications requiring localized and long-term drug delivery and treatment.

A representative study reported a composite incorporating porphyrin into a four-branched PEG–PCL thermosensitive hydrogel (POR–PEG–PCL). This system enabled real-time imaging of drug release through its dual fluorescence properties, and demonstrated effective tumor suppression along with the sustained release of doxorubicin (Dox) [73]. Furthermore, a chitosan-based composite scaffold (alginate/chitosan/PLA-H scaffold) coated with PLA-H loaded with VEGF-containing microspheres achieved controlled release for approximately 5 weeks, and induced angiogenesis and bone regeneration while maintaining over 90% biological activity [74]. Meanwhile, a PLGA–PVA/collagen dual-network hydrogel (PTX–NPs–DN hydrogel) enabled the sustained local release of paclitaxel (PTX) for more than 10 days, reducing systemic toxicity and demonstrating potent antitumor efficacy after surgery [75] (Table 4).

### 4.4. Microspheres and Implants: Sustained Drug Release for Long-Term Therapy

While hydrogels and scaffolds are soft, porous structures that promote tissue regeneration while delivering localized drugs, microspheres and implants are designed with more robust and stable structures and are primarily used to treat various chronic diseases requiring prolonged drug release. These systems are designed as particles or implantable structures sized from a few micrometers to several hundred micrometers in size and are optimized to release drugs slowly and at a constant rate over a very long period of time [76,77].

PCL-based microspheres and implants possess high hydrophobicity and crystallinity, leading to very slow hydrolysis. This allows them to release drugs consistently over very long periods of time, from months to years. Therefore, they are ideal for applications requiring sustained and stable drug release, such as diabetes treatments, contraceptives, and long-term hormone therapy. Furthermore, PCL’s biocompatibility and low inflammatory response provide significant advantages in ensuring safety during long-term implantation [78,79]. PLA-based microspheres and implants exhibit relatively rapid and predictable degradation rates, making them ideal for use as intermediate-duration drug delivery systems. In particular, the high mechanical strength of PLA enhances structural durability when manufactured as implants, enabling long-term drug release without physical damage. This is particularly effective for applications requiring the slow delivery of drugs, such as anticancer drugs or local anesthetics, over weeks or months. The lactic acid byproduct generated after degradation is safely metabolized in vivo [80,81].

PLGA-based microspheres and implants can be precisely tuned to achieve precise drug release over a desired timeframe. PLGA biodegradation occurs rapidly and consistently in vivo, and can be designed to suppress the initial rapid drug release. Therefore, PLGA-based systems can be optimized for drug release for a variety of durations and purposes, and are widely used in therapeutic areas requiring precise drug release, such as anticancer treatment and immunotherapy delivery [82,83]. Microspheres and implants are positioned as innovative solutions for chronic disease management and long-term drug therapy due to their ability to maintain consistent drug concentrations over long periods of time, and their precise design, leveraging the biodegradability and mechanical properties of polymers, has enabled successful application in a variety of therapeutic settings.

A representative study reported doxycycline-loaded PCL microspheres prepared by a single emulsion-solvent evaporation method. This formulation demonstrated sustained antibiotic release for approximately 3 months, and the release rate was primarily controlled by diffusion, which was tunable depending on the molecular weight of PCL (14 kDa vs. 65 kDa) [84]. Furthermore, chitosan-lipid composite implants containing PLA–PEG/PLA nanoparticles locally released paclitaxel (PTX) in ascites for up to 4 weeks, and showed a high correlation between in vitro and in vivo release, demonstrating significant tumor suppression in an ovarian cancer model [85]. Meanwhile, PLA–PEG–PLA microspheres prepared by the O/W solvent evaporation method achieved approximately 50% PTX release for approximately 1 month due to enhanced porosity caused by the introduction of hydrophilic PEG blocks, and exhibited better biocompatibility than PLGA microspheres [86]. Finally, porous PLGA microspheres prepared by W/O/W composite emulsification using ammonium bicarbonate induced synergistic cytotoxicity through dual drug loading of dosorubicin (DOX) and paclitaxel (PTX), effectively reducing systemic toxicity while suggesting an optimal drug ratio (DOX:PTX = 2:1) suitable for inhalational treatment of lung cancer [87] (Table 5).

## 5. Mechanisms of Drug Release and Controlled Delivery

### 5.1. Diffusion, Degradation, and Erosion-Controlled Release

Drug release refers to the process by which a drug moves from its initial location within a drug delivery system to the external environment. It is generally categorized as immediate release and controlled release. Controlled-release systems can maximize therapeutic efficacy and minimize side effects by maintaining a stable release rate over a certain period of time or inducing selective release at specific biological sites [88].

In biodegradable polymer-based systems such as PCL, PLA, and PLGA, drug release is primarily controlled by diffusion, polymer degradation, and erosion mechanisms. Diffusion is based on the movement of a drug from a high concentration to a low concentration according to Fick’s law. Diffusion is dominant in the initial phase, with polymer degradation and erosion gradually taking over over time [89,90].

Due to its high crystallinity and hydrophobicity, PCL restricts water penetration, and its slow hydrolysis rate, rather than diffusion, is the primary limiting factor in the release rate. Gentle bulk erosion dominates over surface erosion, and limited pore formation results in minimal initial release and a relatively stable zero-order release pattern [91]. These characteristics enable PCL-based systems to provide prolonged release for months or even years, making them suitable for formulations requiring sustained, consistent concentrations, such as hormones, contraceptives, and long-term anticancer drugs. Strategies such as copolymerization, surface activation, and hydrophilic filler incorporation are utilized to control the degradation rate [92].

In PLA, diffusion through water-filled pores is the most common release mechanism. PLA exhibits prominent bulk erosion characteristics, resulting in a three-stage release profile: drug diffusion without change in appearance during internal degradation, followed by a rapid increase in release rate in the latter stages as structural collapse begins. The degradation rate and release pattern are controlled by copolymer composition, molecular weight, crystallinity, and hydrophilicity [93,94]. PLGA also exhibits bulk erosion properties, and its hydrophilicity and degradation rate can be precisely controlled, particularly by varying the LA:GA ratio [95]. In matrices such as PLGA hydrogels, swelling and network relaxation due to water absorption significantly affect the release rate [96]. Structural characteristics (e.g., porosity, degree of cross-linking) are key determinants of the release rate. In drug-polymer conjugates, the drug is linked to the polymer chain via ester or amide bonds, and is cleaved and released by external stimuli such as hydrolysis, redox, and enzymatic action [97].

Furthermore, Monte Carlo simulation models can quantitatively predict release time by distinguishing between diffusion-dominated and reaction-dominated regimes, making them particularly useful in systems with limited matrix erosion [98]. As a result, the diffusion, decomposition, and erosion-based release mechanisms show different aspects depending on the physical and chemical properties and structures of PCL, PLA, and PLGA, respectively, and by precisely designing and controlling these, long-term and precise anticancer drug release can be realized.

### 5.2. Stimuli-Responsive Systems (pH-Sensitive, Temperature-Sensitive, Enzyme-Triggered)

Stimuli-responsive polymeric nanocarriers are smart drug delivery systems designed to selectively release drugs in response to specific endogenous stimuli (e.g., pH, enzymes, redox) or exogenous stimuli (e.g., temperature, light, ultrasound, magnetic fields, and electric fields) [99,100]. Nanoparticles based on PCL, PLA, and PLGA can induce precise drug release in response to characteristics of the tumor microenvironment, such as low pH, high glutathione (GSH) concentration, and overexpression of specific enzymes [101].

Ph-sensitive systems utilize acid-cleavable bonds such as hydrazone, imine, and cis-aconityl to release drugs in acidic environments such as endosomes and lysosomes. Polyacrylic acid (PAA), polymethacrylic acid (PMAA), and poly(N,N-dimethylaminoethyl methacrylate, PDMAEMA) are used for this purpose [102]. PLA and PLGA-based systems can be designed to rapidly cleave in acidic environments, while PCL-based systems can be surface-modified and block copolymerized to impart pH-sensitive functional groups for accelerated release [103].

Temperature-responsive systems, when the hydrophobicity of the polymer increases above the lower critical solution temperature (LCST), result in nanostructure collapse and drug release. Representative examples include PNIPAAm, Pluronic (PEO–PPO–PEO block copolymer), gelatin, and chitosan. Copolymers incorporating PCL or PLA blocks can simultaneously exhibit temperature sensitivity and biodegradability [104].

Enzyme-responsive systems are designed to cleave the peptide linker or polymer backbone by enzymes overexpressed in tumor tissues, such as matrix metalloproteinases (MMPs), cathepsin B, and esterases. Examples include PLGLAG sequences, hyaluronidase (HAase)-sensitive hyaluronic acid coatings, and plasmin-responsive protein-based structures. Incorporating these linkers into PLA and PLGA matrices allows for selective release at the tumor site, and PCL-based systems can also be surface functionalized to impart enzyme sensitivity [105,106].

These stimuli-responsive carriers maintain stability in normal tissues while releasing drugs only at the target site, preventing premature leakage and enhancing therapeutic efficacy while minimizing side effects. Furthermore, multimodal platforms that respond to dual or triple stimuli, such as pH/redox, pH/temperature, and light/redox, are also being developed [107]. These advanced systems offer precise control over the timing and rate of drug release depending on the stimulus conditions, and are attracting attention as next-generation anticancer treatment strategies.

### 5.3. Dual and Multi-Modal Drug Delivery Strategies

Dual and multi-modal drug delivery strategies are important approaches that overcome the limitations of single anticancer agents and simultaneously target not only cancer cells but also the tumor microenvironment (TME), thereby enhancing anticancer efficacy and therapeutic precision [108]. Biodegradable polymeric carriers based on PCL, PLA, and PLGA can be designed as theranostic systems that can integrate different therapeutic mechanisms, such as chemotherapy, photothermal therapy (PTT), photodynamic therapy (PDT), gene therapy, and immunotherapy, into a single platform. Incorporating functional nanomaterials such as polydopamine (PDA), black phosphorus (BP), indocyanine green (ICG), gold nanorods (GNR), and porous silicon nanoparticles (PSiNPs) enables simultaneous drug release, imaging diagnosis, and multiple therapeutics within a single formulation [109]. For example, PDA-based ICG-PDA-TPZ nanoparticles generated ROS upon near-infrared (NIR) irradiation, activating the hypoxia-responsive prodrug tirapazamine (TPZ) to induce DNA damage, resulting in trimodal therapeutic effects in combination with PTT–PDT–chemotherapy [110].

PLA and PLGA-based cross-linked nanogels exhibit superior biostability and controlled drug release compared to conventional micelles or liposomes, enabling simultaneous implementation of optical imaging, enzyme-mediated therapy, and chemotherapy [111]. Controlling the composition of PLGA and surface functionalization allows for precise treatment sequences (e.g., chemotherapy followed by PDT or immunotherapy), maximizing therapeutic synergy [112]. PCL-based carriers inherently leverage their slow degradation rate and high crystallinity, making them advantageous for multimodal designs requiring prolonged release (e.g., long-term chemotherapy + intermittent PTT/PDT). Surface modification, incorporating pH-, enzyme-, and photoresponsive materials, allows for controlled release tailored to multiple therapeutic environments [113].

These integrated strategies for polymer-based complex carriers play a crucial role in overcoming tumor heterogeneity and therapeutic resistance, and hold great potential for development into next-generation precision anticancer platforms combining therapeutics, diagnostics, and targeting [114]. In particular, PCL, PLA, and PLGA each offer distinct advantages in terms of in vivo stability, degradation rate, and ease of functionalization. Therefore, it is crucial to select and combine materials tailored to the therapeutic intent and disease characteristics.

## 6. Active and Passive Targeting Strategies for Enhanced Efficacy

### 6.1. Enhanced Permeability and Retention (EPR) Effect for Passive Targeting

Passive targeting is a targeted drug delivery strategy that leverages tumor-specific vascular architecture, based on the enhanced permeability and retention (EPR) effect [115]. Unlike normal tissue, cancer tissue blood vessels are incompletely formed during rapid angiogenesis, resulting in wide endothelial gaps and impaired lymphatic drainage. These structural and functional characteristics allow nanoparticles with diameters ranging from approximately 10–200 nm to selectively penetrate tumor tissue and remain there for extended periods [116].

PCL-based nanoparticles maintain long-term stability in the bloodstream due to their high hydrophobicity and long circulating half-life, enabling prolonged drug release within the tumor microenvironment (TME). These properties are advantageous for the design of anticancer drugs or sustained-release therapeutics requiring prolonged release [117]. PLA-based nanoparticles exhibit relatively rapid hydrolysis rates, allowing them to release drugs within a relatively short period of time after reaching tumor tissue, making them ideal for strategies that rapidly increase local drug concentrations and achieve rapid therapeutic effects [118].

PLGA-based nanoparticles can be precisely engineered to achieve both a degradation rate and release duration by adjusting the lactic acid:glycolic acid (LA:GA) ratio. This allows for stable drug release within tumor tissues while minimizing initial burst release, and maintains consistent concentrations over long periods, enhancing intratumoral therapeutic efficacy [119].

Recent research has actively developed strategies to maximize the EPR effect and enhance tumor selectivity by fine-tuning particle size, surface hydrophilicity/hydrophobicity ratio, and surface charge (zeta potential) [120]. This design optimization directly impacts nanoparticle dynamics (drug circulation, tissue distribution, and tumor penetration) and serves as a key factor in further enhancing the efficiency of passive targeting.

### 6.2. Ligand-Functionalized Nanoparticles for Receptor-Mediated Targeting

Active targeting is a strategy that achieves selective and efficient drug delivery to target cells by attaching ligands that can selectively bind to specific cell receptors to the nanoparticle surface [121].

PCL-based nanoparticles are actively being studied for targeting the αvβ3 integrin receptor by introducing specific ligands, such as cRGD peptides, onto their surface [122]. αvβ3 integrin is a receptor overexpressed on tumor cells and neovascular endothelial cells. After ligand binding, nanoparticles enter cells via receptor-mediated endocytosis, releasing the drug [123].

PLA-based nanoparticles are used by conjugating antibodies (e.g., trastuzumab, cetuximab) that target cancer cell surface receptors such as HER2 and EGFR (epidermal growth factor receptor) to their surface [124]. These antibody-based ligands bind to target receptors and induce endocytosis to deliver drugs into cells, demonstrating particularly effective therapeutic effects in cancers with high receptor expression, such as breast and lung cancer [125].

PLGA-based nanoparticles are effectively utilized for active targeting in the treatment of central nervous system diseases. When polysorbate 80 is attached to the surface, it specifically binds to the low-density lipoprotein (LDL) receptor within the blood–brain barrier (BBB), allowing the nanoparticles to cross the BBB via transcytosis via vascular endothelial cells. This enables efficient delivery of therapeutics for central nervous system diseases such as brain tumors and Alzheimer’s disease [126,127].

Consequently, PCL, PLA, and PLGA-based nanoparticle systems can precisely deliver drugs to target cells by optimizing their individual material properties and ligand-receptor binding strategies. This approach provides an important foundation for the development of customized nano-drug delivery platforms that minimize systemic drug toxicity and maximize therapeutic efficacy.

### 6.3. Multi-Functionalized Polymeric Carriers for Precision Medicine

Multifunctional polymeric carriers are emerging as a key strategy in precision medicine, serving as theranostics platforms that simultaneously deliver therapy and diagnosis. These systems are designed to combine multiple functional modules into a single carrier, enabling concurrent drug release, imaging diagnostics, and environmental stimulus-responsive therapy [128].

PLA-based multifunctional nanoparticles can simultaneously perform real-time diagnostics via magnetic resonance imaging (MRI) and magnetically induced hyperthermia by incorporating magnetic nanoparticles onto their surface. When magnetic nanoparticles are subjected to an external magnetic field, they locally generate heat, damaging cancer cells and simultaneously altering the structure of the PCL matrix, promoting drug release. This heat-release coupling maximizes therapeutic efficacy and enables sustained treatment by leveraging long-term release characteristics [129,130].

PLA-based multifunctional carriers are frequently utilized in designs that combine photodynamic therapy (PDT) with fluorescence imaging. PLA nanoparticles, which incorporate photosensitizers on their surface, are activated by light of a specific wavelength, generating reactive oxygen species (ROS) within the cell and selectively killing cancer cells. Simultaneously, they emit a fluorescent signal, enabling real-time visualization of tumor sites, enabling a truly theranostic approach that combines treatment and diagnosis [131,132].

PLGA-based multifunctional systems can easily incorporate smart functions that respond to environmental stimuli such as pH, temperature, and enzymes. For example, exposure to the acidic environment (lysosomes/endosomes) within the cell accelerates hydrolysis, rapidly releasing drugs. Furthermore, by incorporating thermosensitive polymers or enzyme-sensitive materials onto their surfaces, the release pattern can be actively controlled based on tumor microenvironmental conditions [133,134].

Thus, multifunctional polymer platforms based on PCL, PLA, and PLGA can simultaneously enable precise spatiotemporal control of drug release, real-time imaging diagnosis, and the simultaneous implementation of multiple therapeutic modalities. These characteristics provide the basis for establishing customized treatment strategies based on the patient’s disease status and target characteristics, and are positioned as a key technology for realizing next-generation precision medicine.

## 7. Clinical and Preclinical Applications in Anticancer Therapy

### 7.1. Current Status of FDA-Approved Formulations Based on PCL, PLA, and PLGA

Biodegradable polymers such as polycaprolactone (PCL), polylactic acid (PLA), and poly(lactic-*co*-glycolic acid) (PLGA) have been utilized as the basis for several FDA-approved drug delivery systems in the field of anticancer therapy due to their advantages of biocompatibility and controllable degradation rates [76]. In particular, PLA and PLGA have been applied to the development of long-acting formulations since the late 1980s, opening a new era in anticancer therapy [70]. Representative examples include leuprolide, a GnRH agonist, developed as a PLGA/PLA microsphere injectable (Lupron Depot^®^, North Chicago, IL, USA) and approved by the FDA in 1989, and goserelin, also approved as a biodegradable PLA implant (Zoladex^®^, Cambridge, UK), used for long-term hormone suppression treatment in hormone-dependent prostate cancer and some breast cancers. These formulations provide stable drug release for 1 to 3 months, reducing the number of administrations and improving treatment compliance [135,136]. Additionally, octreotide, a somatostatin analog, has been approved as a PLGA-based once-monthly injectable formulation (Sandostatin^®^ LAR, East Hanover, NJ, USA) for the treatment of neuroendocrine tumors and carcinoid syndrome [137]. Meanwhile, PCL has a relatively slow degradation rate, making it suitable for long-term release implantable formulations, but its application in current anticancer therapeutics is limited compared to PLA and PLGA (Table 6). Overall, these injectable microspheres and implantable formulations have contributed to enhancing therapeutic efficacy by delivering drugs continuously and locally to tumor-related tissues, while reducing rapid fluctuations in blood concentrations and systemic toxicity [76]. Therefore, the clinical success of PCL-, PLA-, and PLGA-based formulations has laid the foundation for next-generation delivery strategies that can be expanded to tumor targeting, combination therapy, and smart formulations, which are closely linked to the advancement of subsequent preclinical studies.

As shown in Table 6, the current status of FDA-approved formulations is summarized to provide an overview of clinically available polymer-based drug delivery systems.

### 7.2. Recent Advancements in Preclinical Research and Animal Studies

Drug delivery systems utilizing biodegradable polymers such as PCL, PLA, and PLGA are being actively studied in various animal tumor models, and have shown promising results in terms of sustained drug release and tumor targeting. For example, in a mouse model study in which arenobufagin was loaded into PEG-PLA (polyethylene glycol-polylactic acid) micelles for tumor-targeted delivery, the tumor inhibition rate was 72.9%, which was 1.28 times higher than that of the free drug, and toxicity was also reduced [138]. Furthermore, when artemisinin (a natural antimalarial agent) was loaded into PLA (polylactic acid) nanoparticles and administered to a rat model of colon cancer, the number and size of tumors were significantly reduced compared to the free drug, demonstrating enhanced anticancer efficacy [139]. Meanwhile, an innovative delivery system for combined chemotherapy and phototherapy was developed by simultaneously loading cisplatin and upconversion nanomaterials onto PLGA nanoparticles surface-modified with RGD peptides, and evaluated in an animal lung cancer model [34]. This RGD-targeted PLGA nanoplatform exhibited antitumor efficacy 4.6 times higher than that of conventional cisplatin injections, along with sustained drug release over approximately 72 h, and also improved treatment safety by minimizing lung tissue damage [140]. In another study, a tumor-targeting ligand (EphA2) was conjugated to microspheres blended with PCL and PLGA, demonstrating excellent therapeutic efficacy by successfully inhibiting local tumor regrowth in triple-negative breast cancer through prolonged drug release for more than 90 days [77]. In particular, such injectable or implantable microsphere formulations have been proposed as a strategy for local, long-term drug delivery to tumor sites while minimizing systemic side effects [77]. In general, PCL, PLA, and PLGA have different degradation rates and physicochemical properties, and these differences are being utilized to precisely control drug release profiles. For example, PCL is slow to degrade, making it suitable for long-term implantable sustained-release formulations, PLGA’s degradation rate can be controlled by varying the lactic acid to glycolic acid ratio, and PLA is widely used as an FDA-approved drug delivery vehicle due to its excellent biocompatibility and mechanical strength [139]. Advances in these preclinical studies suggest that PCL-, PLA-, and PLGA-based systems can significantly enhance therapeutic efficacy in animal tumor models by incorporating innovative formulation strategies such as sustained-release formulations, tumor-targeted delivery (e.g., ligand conjugation, surface modification), and photoactivation systems or implantable depots [141].

The recent advancements in preclinical research and animal studies are summarized in Table 7 to highlight the progress of polymer-based therapeutic systems toward clinical translation.

### 7.3. Challenges in Clinical Translation and Regulatory Considerations

Although PCL, PLA, and PLGA-based drug delivery systems have demonstrated excellent efficacy and safety in preclinical trials, they face technical, biological, and regulatory hurdles during clinical translation [142]. Key technical challenges include the complexity of formulation design, process reproducibility during mass production, maintaining drug stability over long-term release, and minimizing local environmental changes due to degradation products. In particular, ensuring consistency in particle size distribution and release rate during mass production is challenging, and the removal of residual solvents and metal catalysts during the manufacturing process is essential [143]. Biological considerations include heterogeneity in biodistribution, reduced target tissue delivery, heterogeneity in the tumor microenvironment, interactions with the immune system, and potential toxicity due to long-term retention. For example, nanoparticles accumulate in RES organs such as the liver and spleen through protein opsonin binding, reducing target delivery efficiency. Furthermore, local pH changes due to PLA/PLGA degradation products can affect drug stability and tissue response [144]. Regulatory delays include the lack of safety verification of polymer matrices and degradation products, GMP and quality control, long-term stability testing, sterility assurance, and standardized release testing methods [145]. In actual clinical trials, although PLA-based paclitaxel micelles (Genexol-PM^®^) demonstrated anticancer efficacy and low toxicity in phase 2 and 3, standardization of the manufacturing process and securing long-term safety remain challenges [146]. Among PLGA-based formulations, leuprolide sustained-release injection (Lupron Depot^®^) has been approved for phase 3 clinical trials and is being used in clinical settings [147], but development of novel platforms such as OncoGel™ and BIND-014 has been halted due to insufficient release reproducibility and consistency in mass production [148]. A PLGA-based nanovaccine (Precious-01), which is currently under development, also shows potential for enhanced immunogenicity, but thorough regulatory verification of long-term toxicity, immunotoxicity, and degradation product accumulation is required [149]. On the other hand, PCL has a slow degradation rate, making it advantageous for long-term release, but few cases have been clinically implemented as an anticancer drug delivery system [150]. In summary, although biodegradable polymer DDSs have successfully demonstrated clinical efficacy, significant challenges remain in meeting regulatory requirements for manufacturing, stability, and safety.

## 8. Conclusions

### 8.1. Summary of Key Findings

This paper comprehensively reviews the design, mechanism of action, applications, and targeting strategies of biodegradable polymer drug delivery systems based on PCL, PLA, and PLGA. These three polymers can be precisely engineered to tailor their degradation rates and drug release profiles by precisely controlling their physicochemical properties, such as molecular weight, composition, and crystallinity. To achieve this, various synthetic and functionalization strategies, including ring-opening polymerization, block copolymerization, surface modification, and the introduction of functional groups, are utilized. This design flexibility allows the polymers to be engineered into diverse platforms, including micelles, nanoparticles, nanocapsules, hydrogels, and microspheres. Each platform offers unique advantages, including enhanced solubility of poorly soluble drugs, prolonged circulation, local and systemic targeted delivery, and prolonged sustained release. Drug release is controlled by diffusion, polymer degradation/erosion, and stimuli-responsive mechanisms. In particular, stimuli-responsive designs that leverage environmental characteristics of the tumor microenvironment, such as low pH, high reducing power, and overexpression of specific enzymes, are emerging as key to enhancing therapeutic selectivity and efficacy. In terms of targeting strategies, passive targeting utilizing the EPR effect enables selective accumulation in tumor tissue, while active targeting via ligand binding enhances therapeutic efficacy by inducing receptor-mediated intracellular uptake. Furthermore, a multifunctional targeting strategy integrating passive and active targeting with stimulus-responsive release can simultaneously achieve target-site accumulation and selective drug release, minimizing side effects and maximizing therapeutic efficacy. These scientific evidences, along with preclinical and clinical examples, demonstrate that PCL, PLA, and PLGA-based systems represent flexible and powerful platforms for a wide range of therapeutic applications.

### 8.2. Implications for Future Research and Clinical Applications

Drug delivery systems based on PCL, PLA, and PLGA have the potential to play a crucial role in future precision medicine and combination therapy platforms. A key future development direction is to precisely design polymer structures and compositions to precisely control drug release rates, release durations, and targeting characteristics. Applying advanced synthetic strategies, such as block copolymer arrangements, branched/star-shaped structures, and the introduction of functional groups during the molecular design stage, can improve physicochemical stability and in vivo behavior. These strategies can enhance the solubility of poorly soluble drugs, increase target tissue accumulation efficiency, and ensure homogeneity in therapeutic responses. In terms of therapeutic strategies, these systems can be expanded to multimodal delivery systems, such as drug–drug, drug–gene, and drug–photo/thermal therapy, to maximize therapeutic efficacy and mitigate clinical challenges such as resistance and relapse. In particular, responsive designs that leverage endogenous stimuli, such as the low pH and high reducing power of the tumor microenvironment, and the overexpression of specific enzymes, enable selective and precise drug release at the target site, minimizing side effects. These characteristics can also be effectively utilized in combination with immunotherapies or gene therapies. Furthermore, personalized design that reflects the patient’s genetic information, disease characteristics, and tumor microenvironment at each stage can optimize targeting ligands and drug combinations for each individual patient, enabling a precision medicine approach. To facilitate clinical translation, standardized manufacturing processes are essential to ensure consistent quality during mass production, a long-term stability and safety verification system is established, and an approval strategy is established through early consultation with regulatory agencies. Furthermore, internationally standardized release testing methods and quality assessment guidelines should be established to lay the foundation for the successful transition of various research platforms into commercial products. If these scientific and technological advances are combined with regulatory optimization, biodegradable polymer-based drug delivery systems will be able to translate current research findings into clinical practice and fully realize their potential as next-generation precision treatment platforms.

## Figures and Tables

**Figure 1 pharmaceutics-17-01354-f001:**
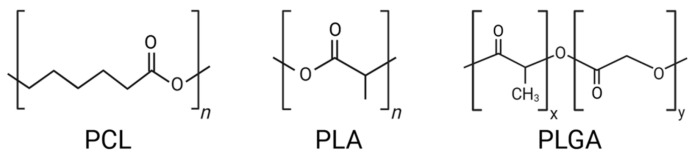
Chemical structures of PCL, PLA, and PLGA.

**Figure 2 pharmaceutics-17-01354-f002:**
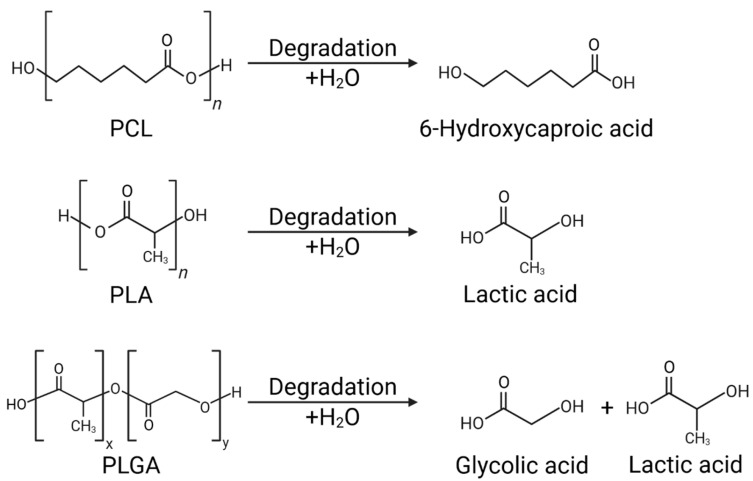
Hydrolysis mechanisms of biodegradable polymers: PCL, PLA, and PLGA.

**Table 2 pharmaceutics-17-01354-t002:** Representative Studies on Polymeric Micelles for Drug Delivery.

Polymer	Formulation	Loaded Drug	Key Findings	Reference
Star-shaped PCL/PEG (SSMPEG-PCL)	Star-shaped micelle prepared by radical polymerization of acrylated MPEG–PCL	Doxorubicin (Dox)	Demonstrated high stability (low CMC), pH-responsive drug release, and superior anticancer efficacy in vitro and in vivo.	[50]
PLA–PEG/ PLA–PEG–Fol	Micellar nanoparticles with or without folate modification	Curcumin (CUR)	Improved solubility and cellular uptake; folate-modified micelles showed enhanced targeting and cytotoxicity toward HepG2 cells.	[51]
PLGA–PEG–PLGA	Triblock copolymeric micelles prepared by solvent–dialysis method	Curcumin (CUR)	Increased drug loading efficiency, extended plasma half-life, and altered biodistribution with reduced RES uptake.	[52]

**Table 3 pharmaceutics-17-01354-t003:** Representative Studies on nanoparticles for Drug Delivery.

Polymer	Formulation	Loaded Drug	Key Findings	Reference
PCL–PEG–PCL triblock copolymer	Self-assembled nanoparticles prepared by ultrasonic emulsion and solvent evaporation	Curcumin (CUR)	High encapsulation efficiency (≈95%), particle size ~60 nm, sustained release up to 96 h, and 4-fold higher AUC and longer circulation time vs. free CUR.	[61]
PEG-PLA (and PEG-PTX hybrid)	PEG-PTX/PTX nanoparticles vs. PEG-PLA/PTX micelles (thin-film hydration method)	Paclitaxel (PTX)	PEG-PTX/PTX showed better stability, higher tumor uptake, stronger antitumor efficacy, and lower toxicity than PEG-PLA/PTX or Taxol^®^.	[62]
PLGA	Paclitaxel-loaded PLGA nanoparticles (nanoprecipitation method)	Paclitaxel (PTX)	Spherical NPs < 200 nm with ~90% drug loading; biphasic release (initial burst + sustained), enhanced cytotoxicity vs. Taxol^®^, suitable for i.v. use.	[63]

**Table 4 pharmaceutics-17-01354-t004:** Representative Studies on Polymeric Hydrogels and Scaffolds for Drug Delivery.

Polymer	Formulation	Loaded Drug/ Application	Key Findings	Reference
PCL (Four-arm PEG–PCL hydrogel)	Thermosensitive porphyrin-incorporated hydrogel (POR–PEG–PCL)	Doxorubicin (Dox)	Dual fluorescent system enabled real-time imaging, sustained release, and effective tumor inhibition.	[73]
PLA (PLA-H coated chitosan scaffold)	Composite alginate/chitosan/PLA-H scaffold containing VEGF-loaded microspheres	VEGF	PLA-H coating allowed controlled VEGF release (~5 weeks), maintained >90% bioactivity, induced angiogenesis and bone regeneration at defect sites.	[74]
PLGA (Double-network hydrogel)	PLGA–PVA/collagen double-network hydrogel (PTX–NPs–DN hydrogel)	Paclitaxel (PTX)	Sustained local release (>10 days), reduced systemic toxicity, and strong antitumor efficacy post-surgery.	[75]

**Table 5 pharmaceutics-17-01354-t005:** Representative Studies on Polymeric Microspheres and Implants for Drug Delivery.

Polymer	Formulation	Loaded Drug/ Application	Key Findings	Reference
PCL	Doxycycline-loaded PCL microspheres (single emulsion–solvent evaporation)	Doxycycline	Sustained antibiotic release for 3 months; release kinetics mainly diffusion-controlled; tunable by PCL molecular weight (14 vs. 65 kDa).	[84]
PLA	Chitosan–lipid implant containing PLA–PEG/PLA nanoparticles	Paclitaxel (PTX)	Localized sustained PTX release up to 4 weeks in ascites fluid; strong correlation between in vitro and in vivo release; effective tumor suppression in ovarian cancer model.	[85]
PLA–PEG–PLA	PLA–PEG–PLA microspheres (O/W solvent evaporation)	Paclitaxel (PTX)	Hydrophilic PEG block improved porosity and drug release (~50% in 1 month); enhanced compatibility vs. PLGA microspheres.	[86]
PLGA	Porous PLGA microspheres (W/O/W double emulsion with ammonium bicarbonate)	Doxorubicin (DOX) + Paclitaxel (PTX)	Dual-drug encapsulation achieved synergistic cytotoxicity; optimal DOX/PTX ratio (2:1) for lung cancer inhalation; reduced systemic toxicity.	[87]

**Table 6 pharmaceutics-17-01354-t006:** FDA-Approved PLA/PLGA-Based Depot Formulations for Cancer Therapy.

Polymer	Formulation Type	Brand Name (Active Ingredient)	FDA Application No. (NDA)	Approval Year	Indication(s)	Key Features	References
PLA/PLGA	Injectable microspheres	Lupron Depot^®^ (Leuprolide)	NDA 019732/S012	1998	Hormone- dependent prostate and breast cancer	Sustained release for 1–3 months, reduced dosing frequency, improved patient compliance	[135]
PLA	Biodegradable implant	Zoladex^®^ (Goserelin)	NDA 019726/S24 & NDA 020578/S3	1998	Hormone- dependent prostate cancer, some breast cancers	Biodegradable implant enabling long-term hormone suppression minimally invasive administration	[136]
PLGA	Long-acting injection (LAR)	Sandostatin^®^ LAR (Octreotide)	NDA 021008	1998	Neuroendocrine tumors, carcinoid syndrome	Monthly injection; stable long-term release	[137]

**Table 7 pharmaceutics-17-01354-t007:** Preclinical Applications of PCL, PLA, and PLGA-Based DDS for Cancer Therapy.

Polymer	Formulation/ Modification	Loaded Drug(s)	Animal Model	Key Findings	References
PEG-PLA micelles	PEG-PLA micelles	Arenobufagin	Mouse tumor model	Tumor inhibition rate 72.9%; 1.28× higher than free drug; reduced toxicity	[138]
PLA nanoparticles	PLA nanoparticles	Artemisinin	Rat colon cancer model	Significant reduction in tumor number and size compared to free drug	[139]
PLGA nanoparticles (RGD-modified with upconversion nanomaterials)	PLGA + RGD peptides, cisplatin + nanomaterials	Cisplatin + UCNPs	Animal lung cancer model	Antitumor efficacy 4.6× higher than cisplatin injection; sustained release ~72 h; reduced lung tissue damage	[140]
PCL/PLGA blended microspheres (EphA2 ligand conjugated)	PCL-PLGA microspheres + EphA2 targeting	Not specified (anticancer drug)	Triple-negative breast cancer (TNBC) model	Inhibited local tumor regrowth; sustained release >90 days; proposed as injectable/implantable depot	[77]

## Data Availability

Not applicable.

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
