# Peer review of "Advances in PCL, PLA, and PLGA-Based Technologies for Anticancer Drug Delivery"

_pharmaceutics, 2025, doi:10.3390/pharmaceutics17101354_

Round 1
Reviewer 1 Report
Comments and Suggestions for Authors
Overall, this is a timely and relevant review on a critical topic in nanomedicine. The manuscript provides a good overview of the properties and applications of PCL, PLA, and PLGA. To further strengthen its impact and utility for the readership, the authors could consider the following points:
- The paper currently discusses PCL, PLA, and PLGA in separate silos. A dedicated section or a summary table directly comparing these three polymers would be immensely valuable, including mechanical properties, hydrolysis rates, key advantages/disadvantages for drug delivery.
- Consider restructuring the manuscript to follow a more problem-oriented approach. For instance, targeting tumors → how these polymers are functionalized for delivering therapuetics.
- The manuscript would benefit from the addition of a graphical abstract or a summary figure. A well-designed visual could effectively synthesize the core themes of the review—comparing PCL, PLA, and PLGA properties, their formulation types, and functionalization strategies—thereby improving accessibility and impact for a broad audience.
Author Response
Comment 1: The paper currently discusses PCL, PLA, and PLGA in separate silos. A dedicated section or a summary table directly comparing these three polymers would be immensely valuable, including mechanical properties, hydrolysis rates, key advantages/disadvantages for drug delivery.
Response 1 : Following a reviewer`s comments, We've added a separate section or summary table directly comparing the three polymers. The contents are as follows.
Comment 2 : Consider restructuring the manuscript to follow a more problem-oriented approach. For instance, targeting tumors → how these polymers are functionalized for delivering therapuetics.
Response 2 : Thank you for your valuable review. This review focused on broadly covering a variety of technologies, intentionally minimizing overly detailed descriptions of individual mechanisms. We believed that excessive mechanism-focused descriptions could actually diminish the paper's thematic focus and readability.
Comment 3 : The manuscript would benefit from the addition of a graphical abstract or a summary figure. A well-designed visual could effectively synthesize the core themes of the review—comparing PCL, PLA, and PLGA properties, their formulation types, and functionalization strategies—thereby improving accessibility and impact for a broad audience.
Response 3 : Following a reviewer`s comments, we produced a graphical abstract. The contents are as follows.

Reviewer 2 Report
Comments and Suggestions for Authors
pharmaceutics-3892407
Advances of PCL, PLA, and PLGA-Based Technologies for Anticancer Drug Delivery
The manuscript by Kim et al. provides an overview of the chemical and physicochemical properties, synthesis strategies, and applications of PCL, PLA, and PLGA in anticancer drug delivery. The topic is relevant and the manuscript is generally well organized. However, several issues should be addressed before it can be considered for publication.
- Abstract: The abstract is overly detailed and should be shortened for conciseness while still capturing the key points of the review.
- Section 1.1 – Background on Polymeric Drug Delivery Systems: This section is too superficial. Please expand it with a more comprehensive introduction to polymer-based drug delivery systems, their advantages, and their relevance in anticancer therapy.
- Introduction: The introduction should be revised to clearly state the motivation for conducting this review, its novelty, and its contribution to the field. The authors should also explain how this work differs from existing reviews on similar topics.
- Section 4: This section is currently underdeveloped. It should be expanded by incorporating findings from a broader range of research articles, including tables to summarize key data and representative figures for better illustration.
- Section 7: In addition to FDA-approved formulations and preclinical studies, the authors should include representative examples of formulations that are currently in clinical trials to provide a more complete perspective.
- Table 2: References should be added to support the data presented in this table.
Author Response
The manuscript by Kim et al. provides an overview of the chemical and physicochemical properties, synthesis strategies, and applications of PCL, PLA, and PLGA in anticancer drug delivery. The topic is relevant and the manuscript is generally well organized. However, several issues should be addressed before it can be considered for publication.
Comment 1 : Abstract : The abstract is overly detailed and should be shortened for conciseness while still capturing the key points of the review.
Response 1 : Following a reviewer`s comments, We made the abstract short by capturing the key points. The contents are as follows.
Comment 2 : Section 1.1 – Background on Polymeric Drug Delivery Systems: This section is too superficial. Please expand it with a more comprehensive introduction to polymer-based drug delivery systems, their advantages, and their relevance in anticancer therapy.
Comment 3: Introduction: The introduction should be revised to clearly state the motivation for conducting this review, its novelty, and its contribution to the field. The authors should also explain how this work differs from existing reviews on similar topics.
Response 2&3 : Following a reviewer`s second and third comments, we revised the content, merging paragraphs into one in the process.
Comment 4 : Section 4: This section is currently underdeveloped. It should be expanded by incorporating findings from a broader range of research articles, including tables to summarize key data and representative figures for better illustration.
Response 4 : Since this review aims to broadly cover a variety of technologies, Chapter 4 focuses on explaining the fundamental concepts and operating principles of each technology. This is intended to help readers understand the overall flow and characteristics of these technologies. This intentional structure was designed to avoid overly detailed mechanisms or individual data points, which could distract from the thematic focus of this paper. Meanwhile, Chapter 7 comprehensively covers data and case studies demonstrating how these technologies have been applied in actual research, ensuring readers have a balanced understanding of both conceptual and applied aspects.
Commnet 5 : Section 7: In addition to FDA-approved formulations and preclinical studies, the authors should include representative examples of formulations that are currently in clinical trials to provide a more complete perspective.
Response 5 : While preclinical studies and FDA approvals for PCL, PLA, and PLGA-based anticancer drug delivery formulations have been reported, the number of representative formulations currently in clinical trials remains limited. This is likely due to barriers such as manufacturing reproducibility, long-term safety, and regulatory compliance. Therefore, in this review, we believe it is more appropriate to expand the discussion to focus on the challenges faced during clinical translation, rather than on specific examples of formulations in clinical trials.
Comment 6 : Table 2: References should be added to support the data presented in this table.
Response 6 : Following a reviewer`s comments, As one table was added, tables 1 and 2 became tables 2 and 3, and we added references to each table. The contents are as follows.

Round 2
Reviewer 2 Report
Comments and Suggestions for Authors
The manuscript was revised in response to the comments. However, as this review focuses on "Advances of PCL, PLA, and PLGA-Based Technologies", the authors should include recent advances in the field. As recommended earlier in comment #4, the authors should expand the manuscript and provide a more in-depth analysis by incorporating findings from a broader range of research articles. This expansion should include tables (similar to Table 3) to summarize key data and representative figures for better illustration.
Author Response
Comment : The manuscript was revised in response to the comments. However, as this review focuses on "Advances of PCL, PLA, and PLGA-Based Technologies", the authors should include recent advances in the field. As recommended earlier in comment #4, the authors should expand the manuscript and provide a more in-depth analysis by incorporating findings from a broader range of research articles. This expansion should include tables (similar to Table 3) to summarize key data and representative figures for better illustration.
Response) Following a reviewer`s comments, We have added tables by examining representative studies.

Round 3
Reviewer 2 Report
Comments and Suggestions for Authors
The manuscript was appropriately revised and can be accepted as is for publication.